


**Evaluation of filtering methods for use on high frequency measurements of**
**landslide displacements**
Sohrab Sharifi[1], Michael T. Hendry[1], Renato Macciotta[1], Trevor Evans[2]
[1]*Department of Civil and Environmental Engineering, University of Alberta, Edmonton, AB,*
*Canada*
[2]*Canadian National Railway, Kamloops, British Columbia, Canada*
**Abstract**
Displacement monitoring is a critical control for risks associated with potentially sudden slope
failures. Instrument measurements are, however, obscured by the presence of scatter. Data
filtering methods aim to reduce the scatter and therefore enhance the performance of early
warning systems (EWSs). The effectiveness of EWSs depends on the lag time between the onset
of acceleration and its detection by the monitoring system, such that a timely warning is issued
for implementation of consequence mitigation strategies. This paper evaluates the performance
of three filtering methods (simple moving average, Gaussian-weighted moving average, and
Savitzky-Golay), and considers their comparative advantages and disadvantages. The evaluation
utilized six levels of randomly generated scatter on synthetic data as well as high-frequency global
navigation satellite system (GNSS) displacement measurements at the Ten-mile landslide in
British Columbia, Canada. The simple moving average method exhibited significant
disadvantages compared to the Gaussian-weighted moving average and Savitzky-Golay
approaches. A framework is presented that can be followed to evaluate the adequacy of different
algorithms for minimizing monitoring data scatter.
**Keywords:** Landslide; Early Warning System; Scatter; Filter; Gaussian-Weighted Moving
Average, Savitzky-Golay


## 1. Introduction

Landslides are associated with significant losses in terms of mortality and financial consequences in countries all over the world. In Canada, landslides have cost Canadians approximately $10 billion since 1841 (Guthrie, 2013) and more than $200 million annually (Clague and Bobrowsky, 2010). Essential infrastructure, such as railways and roads that play vital roles in the Canadian economy, can be exposed to damage as they transverse landslide-prone areas. Attempting to completely prevent landslides is typically not feasible, as stabilizing options and realignment may not be cost-effective nor environmentally friendly. This accentuates the significance of adopting strategies that require constant monitoring to mitigate the consequences of sudden landslide collapses (Vaziri et al., 2010; Macciotta and Hendry, 2021).

In recent years, detailed studies have addressed the use of early warning systems (EWSs) as a robust approach to landslide risk management (Intrieri et al., 2012; Thiebes et al., 2014; Atzeni et al., 2015; Hongtao, 2020). The United Nations defines an EWS as "a chain of capacities to provide adequate warning of imminent failure, such that the community and authorities can act accordingly to minimize the consequences associated with failure" (UNISDR, 2009). Although an EWS comprises various components acting interactively, the core of its performance relies on its ability to detect the magnitude and rate of landslide displacement (Intrieri et al., 2012). Given that the timely response of an EWS determines its effectiveness, an accurate sense of landslide velocity and acceleration is necessary. Monitoring instruments able to provide real-time or near real-time readings such as global navigation satellite systems (GNSS) systems and some remote sensing techniques are satisfactory for this purpose (Yin et al., 2010; Tofani et al., 2013; Benoit et al., 2015; Macciotta et al., 2016; Casagli et al., 2017; Chae et al., 2017; Rodriguez et al., 2017, 2018, 2020; Huntley et al., 2017; Intrieri et al., 2018; Journault et al., 2018; Carlà et al., 2019; Deane, 2020; Woods et al., 2020, 2021). These instruments can record the displacement of locations at the surface of the landslide with high temporal resolution, which allows the monitoring system to track movements on the order of a few millimeters per year. In practice, the results are





usually obscured by the presence of scatter, also known as noise, and outliers that affect the

quality of observations. These unfavorable interferences do not reflect the true behavior of the

ground motion and stem from sources such as the external environment and the quality of the

communication signals and wave propagation in the case of remote sensing techniques (Wang,

2011; Carlà et al., 2017b). Outliers are defined herein as abnormal inconsistencies (e.g.,

displacement directions, magnitudes) when compared to the majority of observations in a random

sampling of data (Zimek and Filzmoser, 2018), whereas scatter is defined as measurement data

distributed around the trend of displacement measurements, such that the average difference

between scatter and the displacement trend is zero and has a defined standard deviation.

Scatter in displacement measurements can significantly impact the evaluation of slope

movements performed on unfiltered data and decrease the reliability of an EWS. This can lead to

false warnings of slope acceleration or unacceptable time lags between the onset of slope failure

and its identification, and therefore a loss of credibility for an EWS (Carlà et al., 2017b; Lacasse

and Nadim, 2009). As a result, scatter should be reduced as much as possible without removing

the true slope displacement trends. This reduction is done by applying algorithms that work as

filters to minimize the amplitude of measured scatter around the displacement trend.

Several approaches have been proposed to filter displacement measurements based on either

the frequency or time domain. Fourier and Wavelet transformations aim to find the frequency

characteristics of the data, then attenuate or amplify certain frequencies. These approaches are

discussed in Karl (1989), who suggests they are not generally appropriate for non-stationary data

such as monitoring data time series. Filters that work on the time domain can be classified as

recursive, kernel, or regression filters. Recursive filters calculate the filtered value at a given time

based on the previous filtered value. An example of a recursive filter is the exponential filtering

function, which can be inferior to other filters that fall under the category of kernel filters (Carlà et

al., 2017b). Kernel filters, which include simple moving average (SMA) and Gaussian-weighted

moving average (GWMA), calculate the filtered values as the weighted average of neighbouring



measurements. Of these two kernel filters, SMA is frequently used in the literature largely due to
its simplicity (Macciotta et al., 2016, 2017b; Carlà et al., 2017a,b, 2018, 2019; Intrieri et al., 2018;
Zhang et al., 2020). Regression filters calculate the filtered values by means of regression
analysis of unfiltered values (e.g., Savitzky-Golay, or S-G) (Savitzky and Golay, 1964; William,
1979; Cleveland, 1981; Cleveland and Devlin, 1988).
This paper presents an approach to detect and remove outliers, evaluates the performance of
three filters—SMA, GWMA, and S-G—, and assesses their suitability to be utilized in an EWS.
The three filters are evaluated against the following criteria: 1) scatter is minimized, 2) true
underlying displacement trends are kept with as little modification as possible, and 3) filtered
displacement trends detect acceleration episodes in a timely manner. Moreover, the paper
investigates the significance of the time lag between a landslide acceleration event and its
identification by a monitoring system for the three filters evaluated.
**2. Methodology**
**2.1. Synthetic Data Generation**
The numerical analysis on synthetic dataset (NASD) approach was adopted, which consists of
synthetic dataset scenarios generated to resemble typical landslide displacement measurements,
including acceleration and deceleration periods. These scenarios are idealizations based on
observations of typical landslide displacements published in the literature (Leroueil, 2001; Intrieri
et al., 2012; Macciotta et al., 2016; Schafer, 2016; Carlà et al., 2017a). A total of 12 dimensionless
scenarios were built, with all data between the coordinates $x=0$, $y=0$ and $x=1$, $y=1$. The $x$
represents time, and normalization between 0 and 1 allows extrapolation of the findings for
variable displacement measurement frequencies (e.g., the full range of $x$ could represent a week,
a month, a year). The analysis of synthetic data was focused on the ability of different algorithms
to minimize scatter and identify changes in measured trends; therefore, $y$ represents any of the
displacement measurement metrics of interest, e.g., displacement, cumulative displacement,
velocity, inverse velocity, etc. Mathematical equations and graphical illustrations of the 12
scenarios are listed in Table 1 and shown in Fig. 1, respectively. Scenarios considered decreasing
trends of *y* from a value of 1 to 0, reflecting cumulative negative displacements or inverse-
velocities; however, it was acknowledged that absolute cumulative displacements and absolute
velocities could show increasing trends. In this regard, the evaluation of synthetic data focused
on timely identification of changes in trends as those associated with accelerating and
decelerating periods, and the results are valid if the scenarios are mirrored to vary from 0 to 1.
Nine of the scenarios are referred to as harmonic scenarios, which are characterized by gradual
changes in the trend of parameter *y.* The remaining three scenarios show sudden variations at or
near *x*=0.5, and are referred to as instantaneous scenarios. Considering the discrete nature of
instrument measurements, and to account for different ranges in measurement frequencies, each
scenario was generated several times, each time with a different number of points (Table *2*).

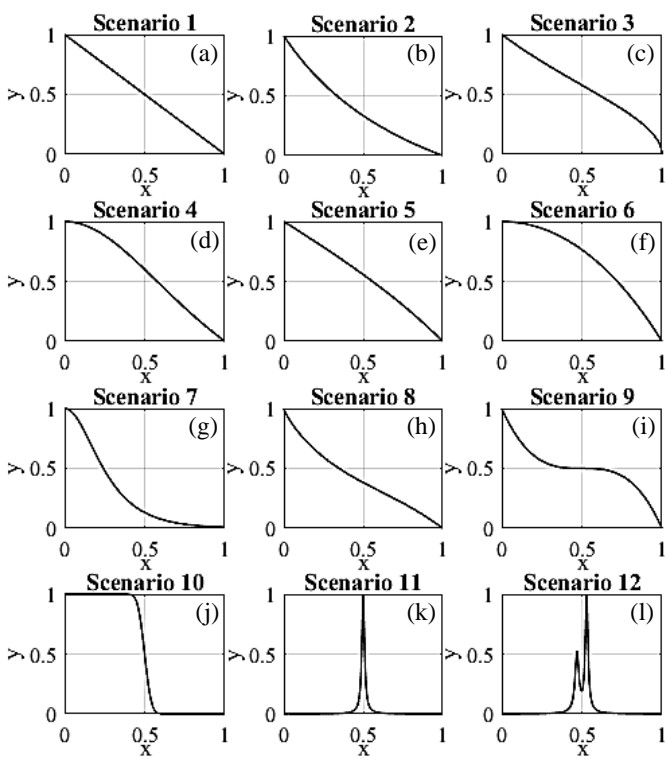

***Fig. 1** Configuration of all synthetically generated scenarios*




**Table 1** Mathematical equations of the 12 generated scenarios

| Scenario No. | Equation |
|---|---|
| 1 | $y=1-x$ |
| 2 | $y=\dfrac{1-x}{1+x}$ |
| 3 | $y=\dfrac{\sqrt{1-x}}{\sqrt{1+x}}$ |
| 4 | $y=\dfrac{1-x^2}{1+x^2}$ |
| 5 | $y=1-\dfrac{e^{-x}-e^x}{e^{-1}-e}$ |
| 6 | $y=1+\dfrac{2-e^{-x}-e^x}{e^{-1}+e-2}$ |
| 7 | $y=\dfrac{2}{e^{2ex}+e^{-2ex}}$ |
| 8 | $y=1+\dfrac{x^{-x}+e^x-2}{1-e}$ |
| 9 | $y=-4(x-0.5)^3+0.5$ |
| 10 | $y=1-0.5\left(1+\text{erf}\left(\dfrac{6x-3}{0.2\sqrt{2}}\right)\right)$ |
| 11 | $y=\dfrac{1}{10^4(x-0.5)^2+1}$ |
| 12 | $y=\dfrac{1}{1.0263}\left[\dfrac{1}{10^4(x-0.47)^2+2}+\dfrac{1}{10^4(x-0.53)^2+1}\right]$ |


**Table 2** Number of points in NASD and examples of their corresponding time spans represented by the
range of *x* from 0 to 1 if the measurement frequency is known (1-h and 60-s readings for illustrative
purposes).

| Number of points | Example monitoring frequency | | | |
|---|---|---|---|---|
| | 1-h readings | | 60-s readings | |
| 1000 | 41.7 | Days | 16.7 | Hours |
| 3000 | 4.1 | Months | 2.1 | Days |
| 9000 | 1.0 | Years | 6.3 | Days |
| 20000 | 2.3 | Years | 2.0 | Weeks |
| 40000 | 4.6 | Years | 4.0 | Weeks |
| 86000 | 9.8 | Years | 2.0 | Months |




| | | | | |
|---|---|---|---|---|
| 250000 | | | 5.8 | Months |
| 500000 | | | 0.9 | Year |
| 750000 | | | 1.4 | Years |
| 1.00E+6 | | | 1.9 | Years |


The next step was adding random scatter to the scenarios to represent unfiltered displacement
measurements. Macciotta et al. (2016) show the scatter in displacement monitoring for a GNSS
system used in their analyses fitted a Gaussian distribution. This was also validated for the data
scatter for the case study in this paper and is presented in a subsequent section. Based on this
observation, the scatter was randomly produced from a normal distribution centred at zero, with
extreme values truncated between −1 and 1 and a standard deviation of 0.20. Random generation
of the scatter followed the techniques outlined in Clifford (1994) known as acceptance-rejection
method, which generates scatter values through a series of iterations until the initial normal
distribution is generated. The amplitude of the scatter around the trend in parameter $y$ was defined
for each scenario based on scaling the randomly generated scatter. This allowed investigation of
the effect of different scatter magnitudes on the performance of the filters. Scaling was done by
defining the ratio $n/t$, which is the ratio of scatter amplitude (maximum deviation around the trend,
termed $n$) to the range of values of the trend ($t$) in each scenario. Six levels of $n/t$ (0.001, 0.005,
0.010, 0.050, 0.100, and 0.150) were considered when performing the analysis to cover a range
of possible levels of scatter in unfiltered measurements. Fig. 2 shows two samples of synthetic
unfiltered scenarios that are the result of superimposing scatter with $n/t$ values of 0.05 and 0.10
on Scenario No. 7.


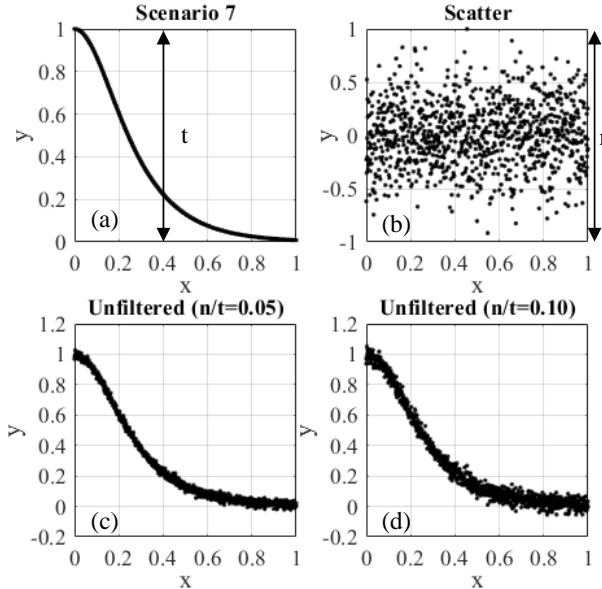

**Fig. 2** *The procedure of generating a scenario with scatter: (a) generated scenario trend, (b) randomly*
*generated scatter, and two scenarios with scatter based on n/t values of (c) 0.05 and (d) 0.10*

### 2.2 Data processing approaches

*2.2.1. Simple moving average*
SMA is a well-known method for scatter reduction that attempts to reduce scatter by calculating
the arithmetic mean of neighbouring points' values. A constant-length interval (window or
bandwidth) is used for the calculation for each point; this is also termed a "running" average.
Equation 1 is the formulation of this method, which was used by Macciotta et al. (2016) to analyze
GNSS data scatter:
$$\hat{y}_i = \frac{\sum_{i-\frac{p-1}{2}}^{i+\frac{p-1}{2}} y_j}{p} ,$$
(1)

where $\hat{y}_i$ is the filtered value, $y_j$ is the unfiltered value, and *p* is the window length. The window
length is constant across the dataset except for the regions near the boundaries as fewer points
are available. Accordingly, *p* will be adjusted to the number of available points that are indeed

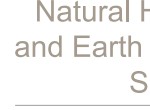

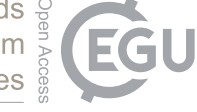

less than the value set by the user. This will cause variation in the effectiveness of the method at
the extremes, which need to be considered when evaluating the results of this approach.
*2.2.2. Gaussian-weighted moving average*
Varying the weights of the measurements within the calculation window in SMA can be used to
develop different filtering methods. The highest weight can be given to the measurement at the
time for which the calculation is being done, with weights decreasing for measurements farther
away in time. One simple weighting function that can be adopted is the Gaussian (normal)
distribution. The filter that assigns weights based on a Gaussian distribution for the averaging
process is:
$$\hat{y}_i = \sum_{i-\frac{p-1}{2}}^{i+\frac{p-1}{2}} w_j y_j \tag{2}$$

where $w_j$ is the weight coefficient based on the Gaussian distribution and the other terms follow
the same definition as per SMA.
*2.2.3. Savitkzy-Golay*
S-G fits a low-degree polynomial equation to the unfiltered measurements within a window and
defines the filtered measurements using the fitted curve (Schafer, 2011). Although this procedure
seems dissimilar from the weighted averaging discussed above, it can be transformed into a
kernel concept using the least-squares method if the data points are evenly spaced. The detailed
procedure is presented in Appendix A. Fig. 3 shows the weight kernel over a window of seven
points attained by fitting a quadratic polynomial. An immediate observation is that some points
are given negative weights.

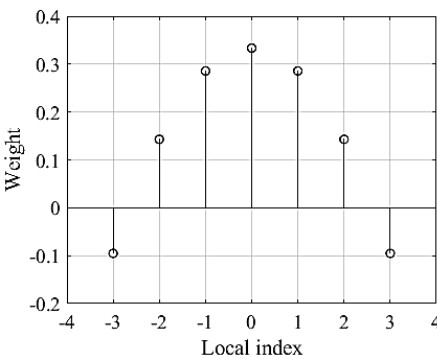


*Fig. 3 The weighting kernel of the Savitzky-Golay filter for seven points*
**2.3 Evaluation of processing algorithms**
The synthetic monitoring data and data from the case studies were filtered using SMA, GWMA,
and S-G techniques. The filters were applied with different lengths of moving windows, from 0.01
(1 %) to 0.1 (10 %) of all monitoring points, referred to as bandwidth ratio (BR). These BR limits
were selected based on literature reports for SMA (Macciotta et al., 2016, 2017b; Carlà et al.,
2017a,b, 2018, 2019; Intrieri et al., 2018; Zhang et al., 2020). Only points prior to the time for
which the calculation is being made are used in the weighted averaging to find the filtered value.
This is to reflect the reality of displacement monitoring information as applied for EWSs. This was
achieved by applying the filters using the time for which the calculation is being made as the
central value, but only utilizing the first half of kernels to assign the weights (the weights are
multiplied by 2 in comparison to a symmetric window to keep the sum of weights equal to 1).
All of these filters require the definition of a bandwidth. A roughness factor was defined to aid in
the evaluation of the effect of bandwidth in reducing scatter. This factor is defined as:
$$J_2 = \frac{\int \hat{y}'' dx}{R_a},$$
(3)

$$R_a = \int y'' dx,$$
(4)



where $J_2$ is the roughness factor, $\hat{y}''$ is the second derivative of filtered measurements, $R_a$ is the
absolute roughness computed by Eq. 4, and $y''$ is the second derivative of unfiltered
measurements. The second derivative measures how much the slope of the line connecting two
consecutive points changes, which itself is an indication of fluctuation. The greater this second
derivative, the greater the variation. $J_2$ was normalized to the overall curvature of the unfiltered
scenario to determine the relative scatter reduction after the application of a filter, eliminating any
roughness associated with the real trend in the scenario. In limit states, a value of 1 means that
fluctuations are similar to the unfiltered dataset, and therefore no improvement has been
achieved; a value of 0 suggests the slope of a scenario remains unchanged and indicates a linear
trend. Because all of the scenarios, except the first, include trends showing concavity or convexity,
a residual value of roughness factor would be expected in the lowest limit state, meaning that a
value of 0 is not necessarily a goal. $J_2$ was used to infer the minimum value for BR after which no
significant change to the fluctuations of results is achieved.
The filters are not expected to remove all scatter, and the error attributed to the residual scatter
can be calculated using the root mean square error (RMSE). Given that velocity values are usually
used as thresholds in an EWS, one concern is whether the filter should be applied to displacement
values or to velocity values derived from unfiltered displacements. To address this issue, two
different approaches to filtering were investigated: direct and indirect. As a result, two different
approaches using the RMSE were also utilized here.
*2.3.1. Direct scatter filtration*
Direct filtration means the filter is applied to the diagram of interest. If the filtered displacement
values are the goal, and the filter is applied to unfiltered displacement values, then the filtering
process is called direct filtration. The same concept applies when velocity values are derived
using unfiltered displacements and the filters are then directly applied to the velocity values. In
this approach, the RMSE follows Eq. 5:





$$RMSEd=\sqrt{\frac{1}{m}\sum_{i=1}^{m}(\hat{y}_i\text{-}y_i)^2},$$
(5)

where *RMSEd* is the measurement of error in direct filtration, $y_i$ is the value of the true trend (for
the synthetic scenario), $\hat{y}_i$ is the filtered value, and *m* is the total number of points. This approach
is often used in the literature (e.g., Macciotta et al., 2016; Carlà et al., 2017a,b, 2018, 2019; Intrieri
et al., 2018).
*2.3.2. Indirect scatter filtration*
Some EWSs can apply the filter to the displacements but use velocity trends as the metric for
evaluation. In this case, the filtered velocity values will be computed using the filtered
displacements. Indirect filtration indicates the diagram of interest is the first derivative of the
diagram to which the filter is applied. The RMSE in this case is defined as:
$$RMSEi=\sqrt{\frac{1}{m}\sum_{i=1}^{m}(\hat{y}_i^{'}\text{-}y_i^{'})^2},$$
(6)

where *RMSEi* is the measurement of error in indirect filtration, $y_i^{'}$ is the first derivative of the true
trend, $\hat{y}_i^{'}$ is the first derivative of filtered data (derived velocity after the filter is applied to the
displacements), and *m* is the total number of points.
**2.4 Lag Quantification**
Only antecedent measurements are fed into the filters, which is expected to result in a lag between
the true trend and when these are identified by the filters. This lag means the calculated value of
velocity or displacement occurred sometime in the past. Consequently, reducing this lag means
less time is lost with respect to providing an early warning. To quantify the induced lag, the filtered
diagrams of all scenarios at all *n/t* ratios and BR values were shifted backwards a number of
points equivalent to 0.001 (0.1 %) to 0.1 (10 %) of all generated points. This is referred to as the
shift ratio (SR). This shift of filtered diagrams is expected to increase their similarity with the true





trend until the best correlation is achieved. The $R^2$ test was used to determine how well the shifted
and filtered results replicate the underlying trend.
**2.5. Geocubes Differential GNSS System**
A Geocubes system is a network of differential GNSS units that works with a single frequency
(1572.42 MHz), making it cost-effective (Dorberstein, 2011; Benoit et al., 2014; Rodriguez et al.,
2018). Geocubes communicate with each other through radio frequency, and a reference unit
outside the boundaries of the landslide is assumed as static for differential correction to increase
the low accuracy associated with single frequency GNSS (Benoit et al., 2014; Rodriguez et al.,
2018). The ability of this system to achieve real-time positioning, remote data collection, and
processing makes it a suitable candidate for incorporation into an EWS. As a result, Geocube
data are used in this study to evaluate the performance of the three mentioned filters.
**2.6. Outlier Detection**
Outlier detection techniques have been proposed based on the statistical characteristics of
datasets. One common example is the Z-score method, which calculates the mean and standard
deviation of data within a defined interval and identifies outlier data as those beyond three
standard deviations from the mean (Rousseeuw and Hubert, 2011). A limitation of this kind of
approach is the sensitivity of the mean and standard deviation to the outlier data points, which
has led to the development of other methods that use other indices such as the median (Salgado
et al., 2016). One such technique that was adopted in this study is the Hampel filter (Hampel,
1971). In this method, the median of the displacement measurements within a running bandwidth
is calculated and data outside a defined threshold from the median are identified as outliers. The
threshold is defined as a constant (threshold factor) multiplied by the median absolute deviation.
An asymmetric window with a bandwidth ratio of 0.004 (0.4%) and a threshold factor of three were
adopted following previous studies (Davies and Gather, 1993; Pearson, 2002; Liu et al., 2004;





Yao et al., 2019). The data identified as outliers were then replaced by linear interpolation of the
displacement measurements.
**3. Study Site – Ten-mile Landslide**
The Ten-mile landslide is located in southwestern British Columbia (BC), in the Fraser River
Valley north of Lillooet (Fig. 4a). It is a reactivated portion of a post-glacial earthflow (Bovis 1985)
that was first recognized in the 1970s. The landslide velocity has increased from an average of 1
mm/day in 2006 to 6 mm/day in 2016, with a maximum measured velocity of 10 mm/day (Gaib et
al., 2012; BGC Engineering Inc., 2016). The movement of this landslide impacts the integrity of
BC Highway 99 and a section of railway operated by Canadian National Railway (CN) (Carlà et
al., 2018), with most movement limited to the volume downslope from the railway due to the
installation of a retaining wall (Macciotta et al., 2017a). Despite the stabilization work done to date,
the uppermost tension crack has retrogressed approximately 200 m in 45 years and is now
situated 60 m upslope of the railway track (Macciotta et al., 2017b). The landslide lateral extents
have not expanded according to the aerial photographs since 1981 (Macciotta et al., 2017b). The
Ten-mile landslide is currently approximately 200 m wide, 140 m high, and has a volume of 0.75
to 1 million m$^3$, moving towards the Fraser River on a continuous rupture surface with a dip of
about 22 to 24°, which is sub-parallel to the ground surface (Rodriguez et al., 2017; Donati et al.,
2020). The elevation of the shear surface and mechanism of the landslide have been inferred
from the readings of multiple slope inclinometers installed in 2015 (BGC Engineering Inc., 2015).
The bedrock in this region consists of volcanic rocks, such as andesite, dacite, and basalt, and is
overlain by Quaternary deposits (Donati et al., 2020; Carlà et al., 2018; Macciotta et al., 2017a).
The thickness of landslide varies between 20 to 40 m and the ground profile from the surface to
depth comprises medium to high plastic clays and silts overlying colluvium material and glacial
deposits, overlying bedrock (BGC Engineering Inc., 2015). The stratigraphy of the sedimented




soils in the landslide area notably varies from one borehole to another, which reflects the complex
stratigraphy of the earthflow.
A total of 11 Geocubes were installed at the Ten-mile landslide in 2016. Fig. 4b is a front view of
the landslide showing the locations of the Geocube units. Units 44 and 50 are installed near the
uppermost tension crack identified as the current landslide backscarp, unit 69 is 30 m above the
backscarp, and unit 39 is used as the reference point. Please note that unit 69 is used for
monitoring for potential retrogression, and is not shown in Fig. 4b. The other units are located
within the boundaries of the landslide, with a maximum distance between units of 310 m
(Rodriguez et al., 2018). The time step between every two consecutive measurements is 60 s.
Fig. 5 shows the displacement of units 46 and 47, which had the largest displacements in
comparison to other Geocubes.

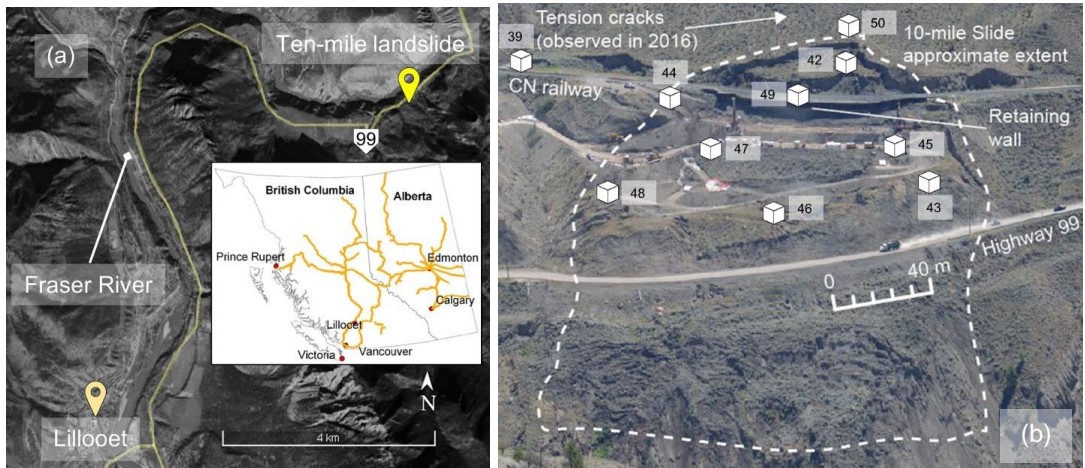

**Fig. 4** *(a) Location of the Ten-mile landslide (base imagery © Google Earth) and (b) front view of the Ten-*
*mile landslide and distribution of Geocubes on its surface (Rodriguez et al., 2018; Macciotta et al., 2017b)*



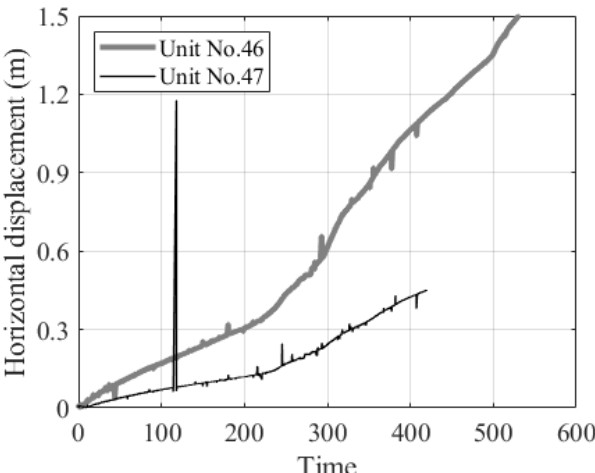

**Fig. 5** *Cumulative horizontal displacement of Geocube units No. 46 and 47*
**4. Results and Discussion**
**4.1. Synthetic Analysis**
Fig. 6 shows the roughness value ($J_2$) of Scenario 6 for SMA, GWMA, and S-G on a semi-
logarithmic scale. This figure illustrates how, regardless of $n/t$ ratio, $J_2$ substantially decreases as
the BR increases to 0.01 and then asymptotically approaches a final value. This means that
increasing the BR drastically reduces scatter; however, its effectiveness is restricted as the BR
increases above 0.01. This observation was consistent for other scenarios. $J_2$ values (including
Scenario 6 in Fig. 6) indicate that $J_2$ approaches its minimum at a BR value of 0.03 to 0.04,
regardless of the filter selected.

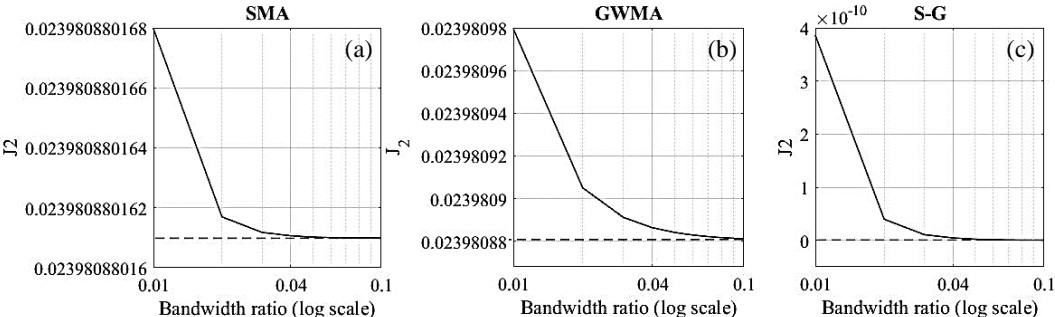


**Fig. 6** *Variation of roughness factor with respect to BR and the applied filter*

Fig. 7 shows the RMSEd of all three filters for all of the harmonic synthetic scenarios. This figure
shows that, for the NASD, the error depends linearly on the BR for all of the filters and does not
depend on the scenario or $n/t$ ratio. SMA shows the greatest difference from the true trend,
followed by GWMA (approximately 60% less difference than SMA). S-G, on the other hand,
almost lies on the horizontal axis for all of the BRs, which means the filtered results yield near
zero error. Fig. 7 also shows how error increases as BR increases. This can be attributed to the
fact that an asymmetric window was utilized, which leads to a lagged response of the filter. As
more points are included in the filtering procedure (increasing BR), this lag increases and,
consequently, causes higher error. The RMSEd of filters for the instantaneous synthetic scenarios
are shown in Fig. 8. In Scenario 10, the same behaviour as for the harmonic scenarios can be
seen from SMA and GWMA, whereas S-G is not as accurate. This is more noticeable in Scenarios
11 and 12 in which S-G becomes less accurate than GWMA at high BRs. This result shows that
S-G cannot handle the instantaneous scenarios as satisfactorily as it does the harmonic ones.
The errors related to SMA and GWMA for the instantaneous synthetic scenarios show non-linear
behavior, and are greater when compared to the harmonic scenarios. Fig. 8 clearly shows all
filters are challenged by the instantaneous variations when compared to gradual ones in direct
filtration.

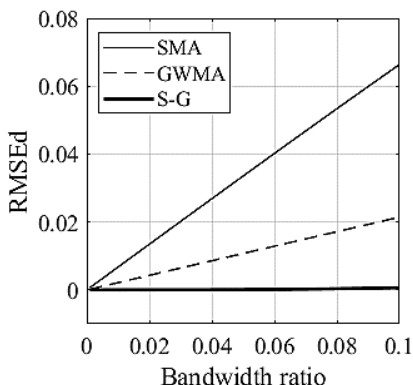

**Fig. 7** *RMSEd for the harmonic scenarios*

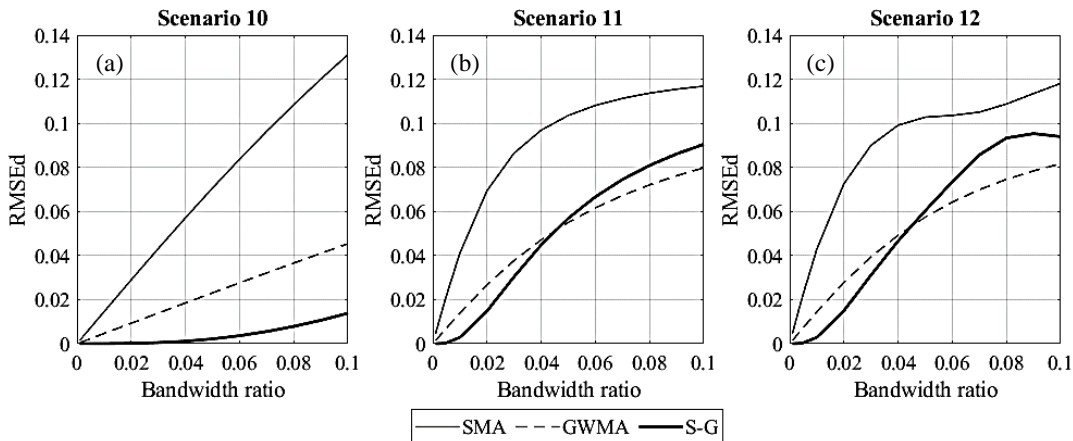

**Fig. 8** *RMSEd for the instantaneous scenarios*

Fig. 9 shows the RMSEi results for the harmonic scenarios (when performing indirect filtration).

The results show the error considerably reduced as the BR increases to 0.01 for SMA and GWMA

and 0.02 for S-G, and has an asymptotic tendency above these BR values. S-G has the highest

error at low BR values in comparison to SMA and GWMA, but shows the least error at BRs above

0.01. At BR values over 0.03, fluctuations do not vary significantly with BR (Fig. 6). In this range

of BR values, the error of GWMA is either equal to or slightly less than the error of SMA, and S-

G shows the least error. The RMSEi results for the instantaneous scenarios (Fig. 10) are similar

to those for the harmonic scenarios for high $n/t$ ratios (0.05, 0.10 and 0.15). For low $n/t$ ratios, the

GWMA is superior at BRs above 0.06, and S-G has the worst performance.



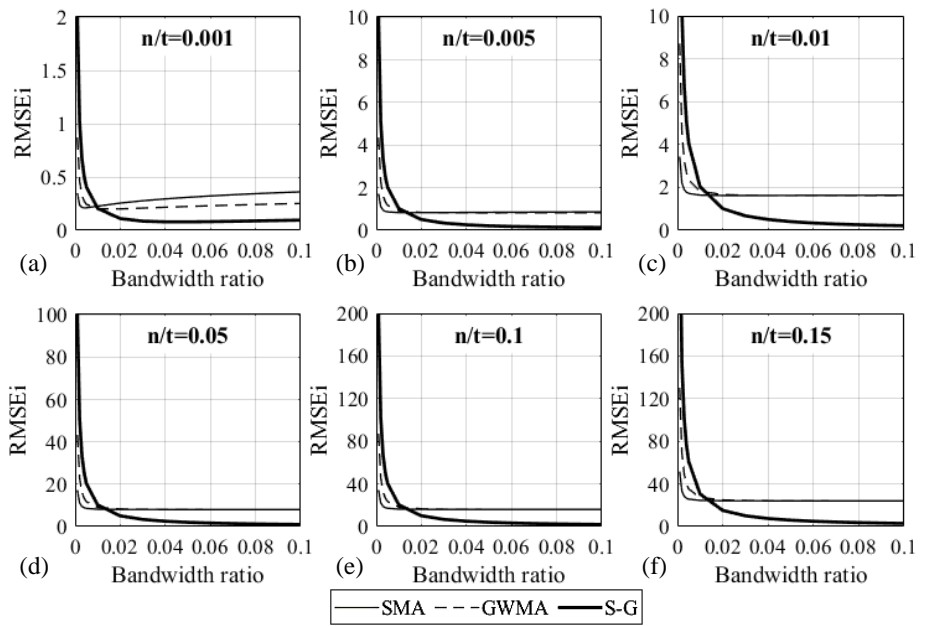

**Fig. 9** *RMSEi for the harmonic scenarios*

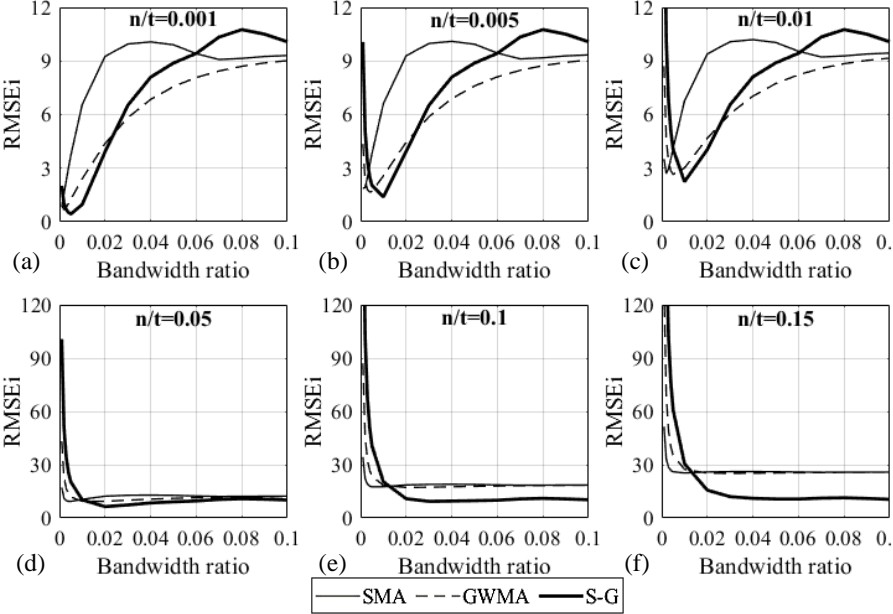

**Fig. 10** *RMSEi for the instantaneous scenarios*

Scenarios 11 and 12 were further analyzed to evaluate how the filter performance is affected by the presence of sudden peak(s). Fig. 11a shows the true trend of Scenario 11 along with two SMA-filtered scenarios at BRs of 0.04 and 0.10. This figure shows that, as the SMA filter

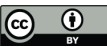
bandwidth increases, the peak in measurements is identified at a later time than the true trend (x
= 0.5) and the magnitude of the peak is reduced (more than 70% reduction at BR=0.10).
Furthermore, as BR increases, the "instantaneous" nature of the peak is lost to a more transitional
variation. This highlights the disadvantage of SMA when handling sudden changes in
displacement trends. The calculated *x* value of the peak in Scenario 11 is plotted for different BR
and for all three filters in Fig. 11b. This figure shows the time at which the peak is identified lags
as the BR increases for all filters; however, GWMA and S-G identify the peak within a much
smaller lag, independent of the *n/t* ratio. As an example, for a year of monitoring data at a
frequency of 30 s and BR=0.10, SMA, GWMA, and S-G predict the peak point approximately 17,
3.5, and 2.7 days after the real peak, respectively. Fig. 11c shows the variation of the peak
magnitude with respect to BR for all three filters. Both SMA and GWMA underestimate the peak
value, and the difference between the calculated peak and real peak increases as BR increases.
SMA calculations underestimate the peak more than twice as much as GWMA. On the contrary,
S-G intensifies the peak up to BR=0.04, with the impact tending to diminish for higher BR values;
it predicts the true value at a BR value of almost 0.09.

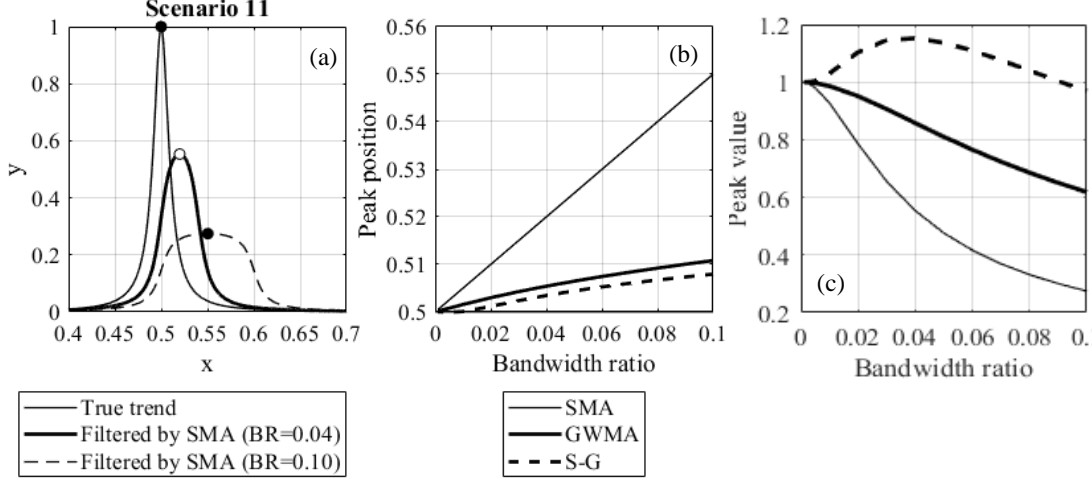

**Fig. 11** *(a) An example of peak displacement by applying SMA, and variation of (b) peak position and (c)*
*peak value with respect to the filter and BR used (original peak at 0.5)*



Scenario 12 was used for a detailed evaluation of the performance of these filters to conserve the
underlying original trend. Fig. 12 shows Scenario 12 and the filtered results for all three filters and
for an $n/t$ ratio of 0.15. This scenario and parameters were selected for illustration purposes as
they allow visual identification of differences for discussion. BR values of 0.04 and 0.10 were
selected as minimum and maximum values after which the scenario had achieved the least error
(lowest RMSEi). The SMA filter considerably underestimates the magnitude of the peak even at
BR=0.04, which is the minimum BR value. At BR=0.10, the filtered diagram is distorted in
comparison to the true trend and the initial peak is not identified. GWMA at a BR of 0.04 shows
less underestimation of the peak magnitude, and a slight lag is visually observed at BR=0.10.
This indicates the significantly better performance of GWMA over SMA. S-G results for both BR
values closely identify the time and magnitude of both peaks, indicating yet better performance.
However, the peak is artificially intensified at BR=0.04, and a significant drop occurs well beyond
the true trend immediately after the second peak for both BR values (pulsating effect), which was
also observed in Scenario 11. Increasing the degree of the polynomial fitted as part of the S-G
methodology was not effective at eliminating this effect. The pulsating effect was also observed
when a symmetrical window was utilized and is attributed to the negative weights in the S-G
kernel.



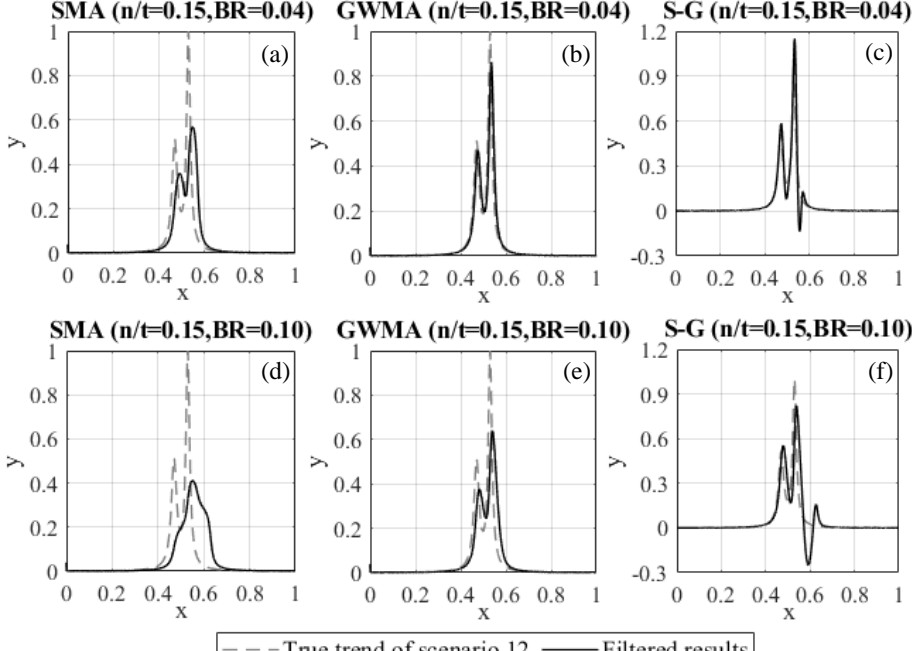

***Fig. 12*** *Filtered results of Scenario 12 with scatter using SMA, GWMA, and S-G at BRs of 0.04 and 0.10*
The lag in identification of monitored trend variations is caused by the non-symmetric inclusion of
points as new information becomes available. Fig. 13 shows Scenario 10 with respect to the
original trend, with scatter added (at $n/t$=0.15), and the results after filtering with each of the three
methods at BR=0.04. This figure clearly shows the lag between the results filtered by SMA and
GWMA and the true trend. S-G results do not have as severe a lag as that resulting from the other
filters; this is attributed to the negative weights in its kernel that anchor the filtered values and
prevent a lagged response. A minor pulsating effect can be observed in the S-G filtered data,
decreasing the calculated values at a much earlier time than the true trend. This suggests that S-
G is robust with respect to identifying initial changes in monitoring trends but overcorrects
subsequent changes; SMA grossly lags with respect to the identification of any change; and
GWMA has a reduced lag when compared to SMA.



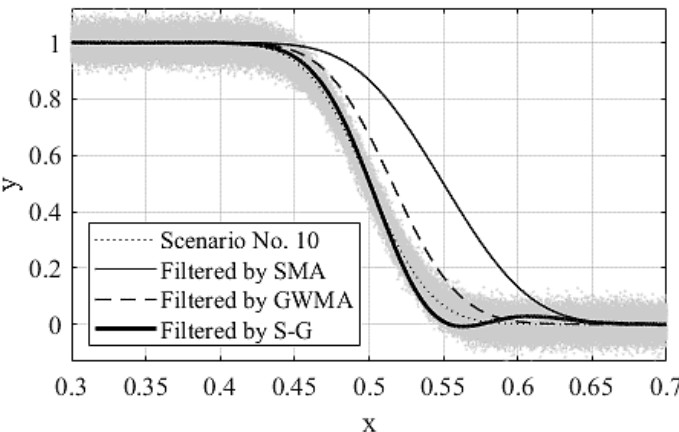

***Fig. 13*** *Scenario 10 with and without scatter, and with scattered results filtered by SMA, GWMA, and S-G*
*for n/t = 0.15 and BR = 0.04.*
Fig. 14a shows an example of $R^2$ correlation for Scenario 7, comparing the original trend and the
results filtered by SMA at n/t = 0.01 and BR = 0.04. SR is the shift of filtered trends (in the
horizontal axis – parameter *x*) relative to the range of *x* values. $R^2$ calculations are shown for the
filtered data (SR=0) and as the filtered trends are shifted backwards in time (negative values of
SR). In this analysis, the peak $R^2$ value (highest correlation between the shifted filtered results
and original trend) indicates the shift required to minimize the lag in identifying the original trend
changes, therefore providing a quantitative approach to calculating the lag in parameter *x*. In the
example in Fig. 14a, the lag corresponded to 0.018 (1.8 %) of the total points.

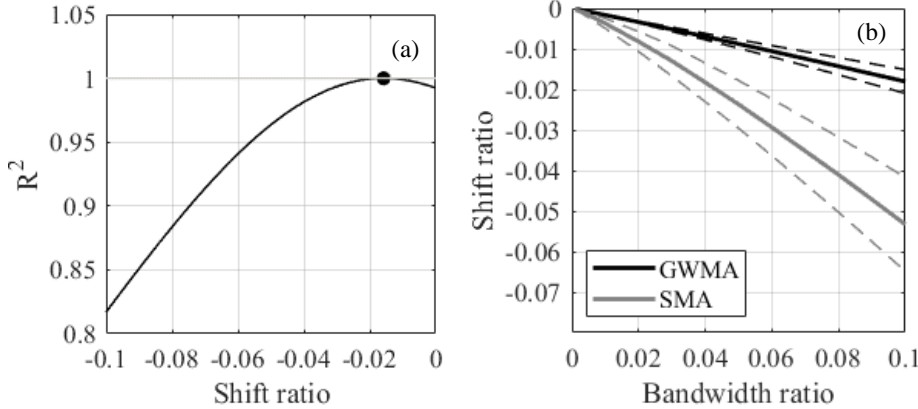

***Fig. 14*** *(a) $R^2$ correlation of Scenario 7 with filtered and shifted results at n/t=0.01 and BR=0.04, (b) shift*
*ratio at peak $R^2$ for all scenarios and n/t ratios, with the mean (solid line) bounded by one standard*
*deviation (dashed lines)*





Peak $R^2$ values for all scenarios and $n/t$ values are closely correlated with the BR. The lag,
quantified by the SR, is higher when the trend change is more pronounced; therefore, the
correlation between SR and BR is different for different scenarios. Fig. 14b shows the mean
correlation between the SR and BR, for all scenarios and n/t values, bounded by one standard
deviation, for GWMA and SMA. Table 3 shows linear and quadratic regressions of this correlation
and the strength of the correlation in terms of $R^2$ and RMSE. Fig. 14b shows quantitatively that
GWMA lags less than SMA with respect to identifying changes in measurement trends. Moreover,
the uncertainty associated with lag in SMA is greater than in GWMA because of larger standard
deviation. Fig. 14b quantifies how increasing BR values increases the lag with respect to
identifying true measurement trends, and although high BR values decrease the scatter in data,
the BR should carefully balance minimizing both scatter ($J_2$) and lag (SR). S-G is not included in
this analysis as the method provided no significant lag in identifying changes in measurement
trends; however, it had the disadvantages previously noted including pulsating effects and
overestimating peak values.
**Table 3** Regression correlations between shift ratio (SR) and bandwidth ratio (BR) with the strength of the
correlation in terms of $R^2$ and RMSE

|  | Linear regression | | Quadratic regression | |
|---|---|---|---|---|
| SMA | SR=-0.5087(BR) | $R^2$=0.9940 RMSE=0.0014 | SR=-1.323$(BR^2)$-0.4049(BR) | $R^2$=0.9997 RMSE=3.24E-4 |
| GWMA | SR=-0.1783(BR) | $R^2$=0.9996 RMSE=1.2963E-4 | SR=-0.1171$(BR^2)$-0.1691(BR) | $R^2$=0.9999 RMSE=3.5672E-5 |

**4.2. Results on the Ten-mile landslide**
Unfiltered results reported by Geocubes 46 and 47 installed on the Ten-mile landslide were
processed by all three filters. To illustrate to the reader through visual inspection the difference
between the performance of SMA, GWMA, and S-G, only a window of 200-day displacement data



of Geocube 46 and filtered points produced by direct filtration are shown in Fig. 15. Although
increasing the BR continues to reduce scatter, it increases the lag in the filtered results, which is
consistent with observations on the synthetic datasets. For BR values over 0.04, SMA becomes
insensitive to some short-scale (20- to 30-day) trends in the data (qualitative visual inspection).
As an example, at BR=0.10, SMA suggests the displacement of Geocube 46 follows a bi-linear
trend with an inflection point at day 240, while unfiltered points and other filters suggest other
periods of acceleration and deceleration. Importantly, S-G is sensitive to even subtle variation
and does not show significant lag.
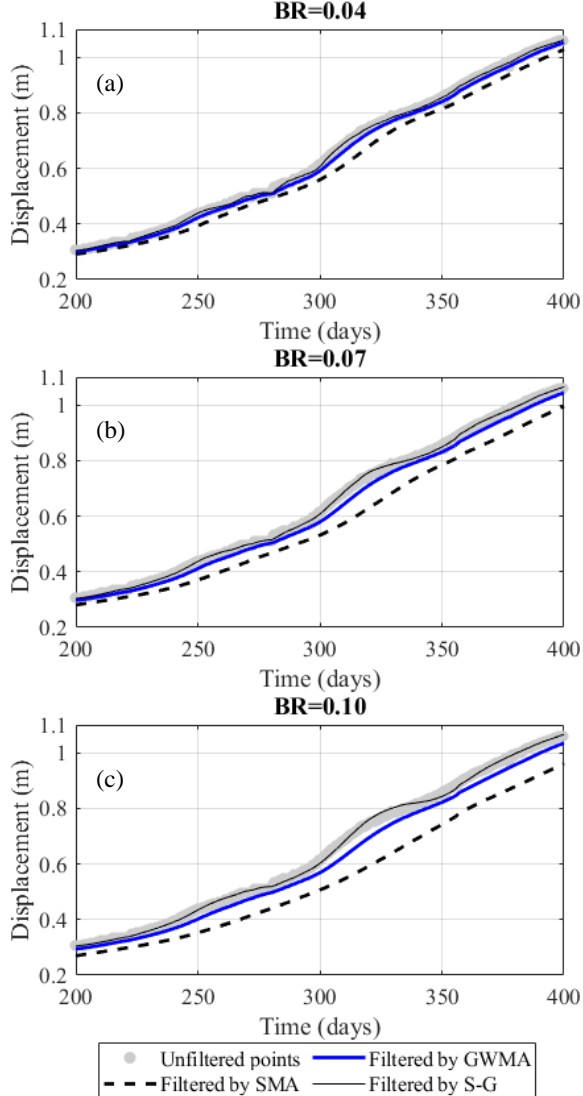

**Fig. 15** *Unfiltered displacement of Geocube 46 vs. time and data filtered by SMA, GWMA, and S-G for different BR values.*

Fig. 16 shows the filtered velocity values obtained by directly filtering the calculated velocities and

by indirectly filtering the displacement values before calculating the velocity for Geocube 46. The

direct and indirect filtering approaches had a similar performance in terms of scatter reduction for

Geocube 46. As the BR increases, SMA tends to significantly attenuate the local maximum and

minimum points in comparison to results at lower BR values, indicating a probable loss of

information about the landslide behaviour and sensitivity of this filter to the BR. Indirect filtration



by SMA seems to be limited near the boundary at time zero, resulting in a subdued replica of

direct filtration. The length of this region is found to be governed by the BR value, as the necessary

number of points for filtering in this portion has not been provided to the filter. This was not

identified as a problem in GWMA, as direct and indirect filtration both follow the same pattern.

Results for Geocube 47 confirm these observations and allow for an evaluation of the significance

of outliers on the filtered results. Fig. 17 shows a magnified portion of the displacement

measurements for Geocube 47 filtered by each of the three filters at three different BRs before

the elimination of outliers. This figure shows that detecting and removing outliers significantly

impacts the performance of S-G, as the presence of the outlier generates a peak that follows the

outlier measurement and is followed by a sudden decrease that goes well beyond the data trend.

SMA tends to widen the range affected by the outlier more than GWMA but, for most part, the

filtered results are almost parallel to the underlying trend. All filters appear to be significantly

impacted by the outlier value, suggesting a pre-processing filter is required to remove outliers

regardless of the use of SMA, GWMA, or S-G to reduce scatter. The outliers were successfully

identified and removed after application of the Hampel algorithm, and the above-mentioned

effects were no longer observed in the filtered results.


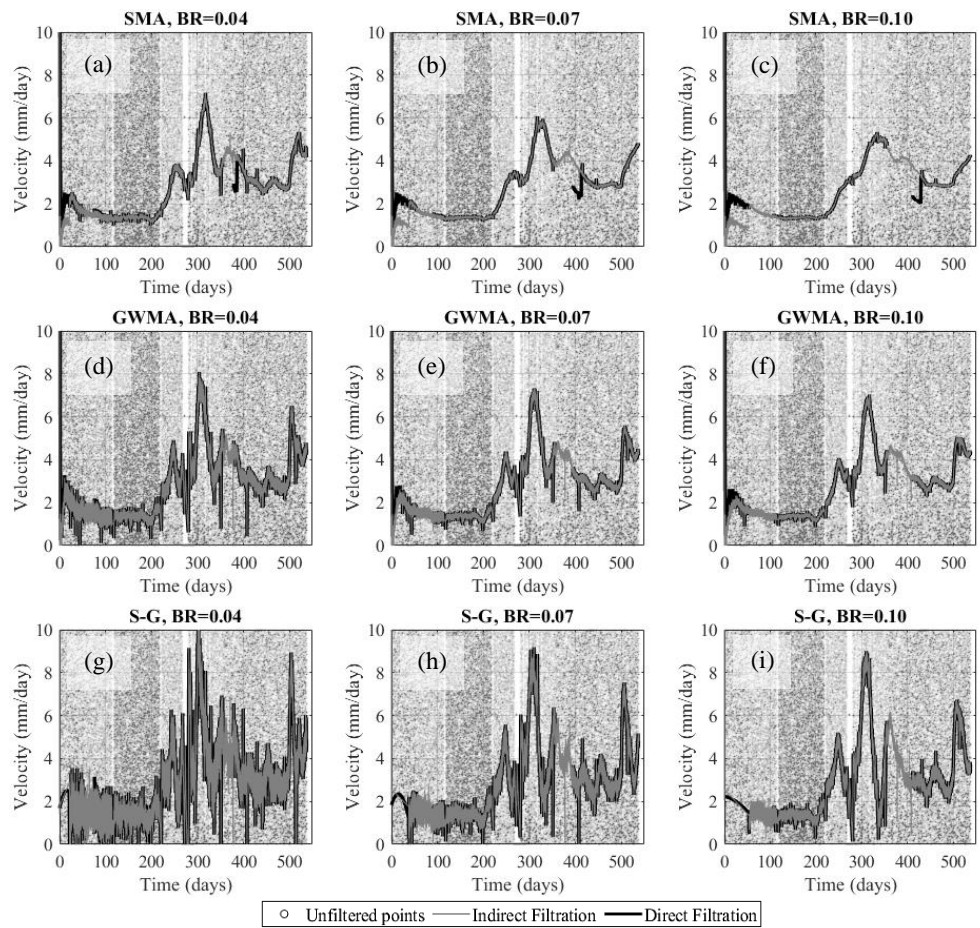

**Fig. 16** *Indirect and direct filtration results of Geocube No. 46 velocity values for BR = 0.04, 0.07, and 0.1.*

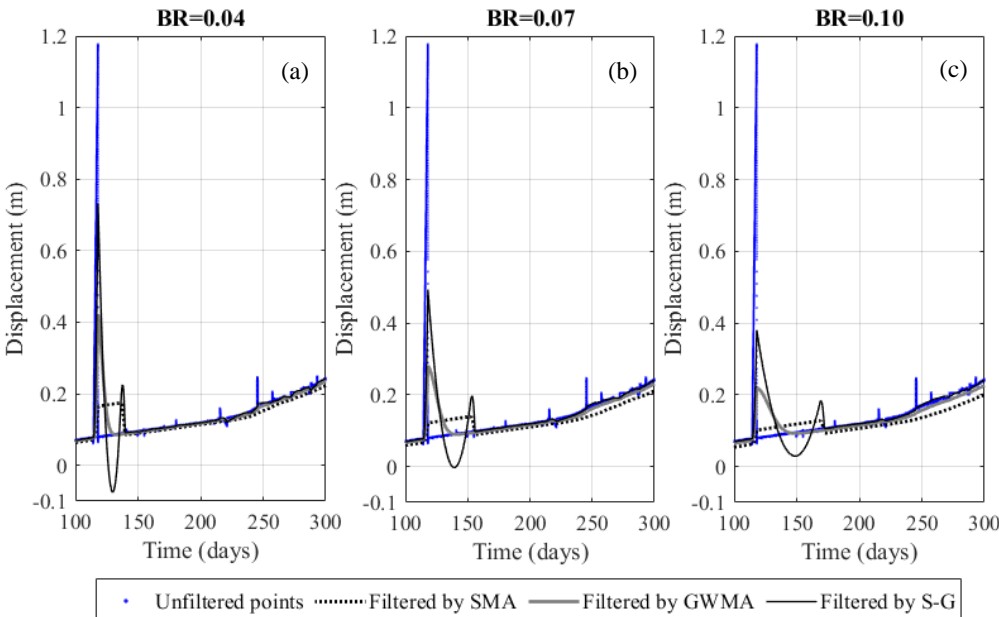

**Fig. 17** *Unfiltered and filtered displacement measurements for Geocube 47 for BR values of 0.04, 0.07 and 0.10*

The lag between unfiltered and filtered data for Geocube 46 (Fig. 15) is consistent with NASD results. The NASD lag quantification results (Fig. 14b and Table 3) were used to provide a correction value for the filtered Geocube results. To determine whether the results of lag correction using the mean correlations derived from NASD (Table 3) were acceptable, the filtered diagrams were shifted (using mean line for GWMA and values between mean and lower boundary for SMA) and different portions of displacement diagrams of Geocubes 46 and 47 were examined. Some examples are tabulated in Table 4. The mean and standard deviation of the scatter around the trend (error distribution) were calculated by assuming a linear trend within the short time periods of analysis in Table 4 (considered an approximation of the true displacement trend for the short time interval). These were also calculated for the filtered and shifted diagrams. The closer the mean and standard deviation of the filtered and shifted data are to that obtained from the linear trend, the better the performance of the lag correction based on NASD results. As an example, for the time period of 250-260 days, the GWMA showed standard deviation of 0.001 to 0.0015 for BR from 0.04 to 0.10, respectively as opposed to 0.0018 to 0.0021 for SMA. This



illustrates that shifted GWMA results are closer to the true (scatter-free) displacements as the
standard deviations of scatter inferred by this filter are closer to the true scatter, although both are
in good agreement with the true scatter. The mean of inferred scatter by both filters are also close
enough to the true scatter's (almost zero). The results show the statistical indices of scatter
inferred from the filtered shifted displacement measurements closely agrees with that considered
to be true scatter, and therefore the filtered displacement measurements are corrected for lag.
This suggests the correlations in Fig. 14b and Table 3 based on NASD are applicable to minimize
the lag for the Geocube system at the Ten-mile landslide.
**Table 4** Mean (unit: m) and standard deviation (unit: m) of scatter inferred by SMA and GWMA in
comparison with true scatter in the displacement of Geocube 46

| Filter | | | SMA | | | GWMA | | | True Scatter |
|---|---|---|---|---|---|---|---|---|---|
| BR | | | 0.04 | 0.07 | 0.10 | 0.04 | 0.07 | 0.10 | |
| Time Period (day) | 60-90 | Mean | -0.0015 | -2.01E-4 | 0.0018 | 0.0010 | 8.86E-4 | 0.0015 | -6.52E-16 |
| | | Std. Dev. | 0.0012 | 0.0012 | 0.0012 | 0.0012 | 0.0012 | 0.0012 | 0.0012 |
| | 250-260 | Mean | -0.0042 | -0.0026 | 0.0010 | 0.0018 | 0.0012 | 0.0012 | 1.17E-6 |
| | | Std. Dev. | 0.0021 | 0.0018 | 0.0018 | 0.0010 | 0.0013 | 0.0015 | 0.0010 |
| | 380-400 | Mean | -0.0048 | -0.0030 | 8.83E-4 | 0.0023 | 0.0017 | 0.0025 | -4.62E-15 |
| | | Std. Dev. | 0.0015 | 0.0014 | 0.0014 | 0.0013 | 0.0013 | 0.0012 | 0.0015 |
| | 410-430 | Mean | -0.0036 | -0.0014 | 0.0026 | 0.0019 | 0.0015 | 0.0025 | 9.91E-16 |
| | | Std. Dev. | 8.80E-4 | 9.30E-4 | 9.61E-4 | 8.32E-4 | 8.24E-4 | 8.33E-4 | 9.42E-4 |







**5. Conclusion**
This study evaluated the suitability of SMA, GWMA, and S-G filters for scatter reduction of
datasets targeted for use in an EWS. A total of different 12 scenarios with harmonic and
instantaneous changes were synthetically generated and random variations with Gaussian
distribution then added to produce unfiltered results. The three filters considered were then each
applied with different bandwidths and the error computed. These filters were also successfully
applied to the records from two Geocubes installed on the Ten-mile landslide. The results led to
the following conclusions:
•   When used for direct filtration of harmonic scenarios, the error resulting from the GWMA
approach was approximately one-third that of the SMA approach. The S-G approach
resulted in near zero error regardless of the BR and $n/t$. When used for direct filtration of
instantaneous scenarios, the superiority of S-G is no longer unconditional and depends
on the BR; this reflects the fact that S-G cannot appropriately handle peaks in the velocity
diagram.
•   When used for indirect filtration of harmonic scenarios, S-G again outperforms the other
methods. The error associated with GWMA is marginally less than for SMA. These
observations are not valid when the filters are applied to instantaneous scenarios, as
GWMA results in less errors than S-G at BRs above 0.03.
•   Detailed investigations with Scenarios 11 and 12 demonstrated that SMA distorts the
underlying trend by displacing and sometimes neglecting peak(s), while GWMA and S-G
tend to preserve them somewhat similarly.
•   Due to the presence of negative weights in the S-G kernel, some artificial smaller troughs
and peaks are created after major peaks. This phenomenon, referred to as pulsating effect
here, results in unfavorable performance of S-G on the velocity and displacement
diagrams, especially in the presence of outliers.





• Investigations on the roughness factor reveal the BR should be at least 0.04. Taking this

into account, GWMA seems to be the most reasonable option as the related uncertainties

are much lower than for S-G and the error is acceptably less than for SMA.

• A consequence of using asymmetric windows in the filtering process is a lag in the SMA

and GWMA results that increases with increasing BR. Lag quantification suggested a

correlation between the needed shift and BR that can be used to eliminate the lag. SMA

requires approximately three times the shift of GWMA on average.

• Application of these filters to displacement data reported by Geocubes illustrates that SMA

and S-G are unable to properly handle data points at the beginning of the dataset (i.e.,

near the boundary) in indirect filtration of the velocity diagram. Moreover, SMA and S-G

are inclined to respectively understate and overstate peaks and fluctuations in the velocity

diagram. Overall, GWMA provides the most reliable filtered values for velocity with no

distinct difference between direct and indirect filtration.

**Appendix A**
Consider a polynomial of degree $k$ that is intended to be fitted over an odd number of points
denoted as $z$. The weighting coefficients of the Savitzky-Golay filter can be extracted from the first
row of matrix $C$ (Eq. 7):
$$C=\left(J^{T}J\right)^{-1}J^{T}, \tag{7}$$
where $T$ operator is the transpose of a matrix and $J$ is the Vandermonde matrix, with elements at
the $i$th row and $j$th column ($1 \leq i \leq z$ and $1 \leq j \leq k+1$) that can be achieved as follows:
$$J_{ij}=m_{i}^{j-1}, \tag{8}$$
where $m$ is the local index of points ($-(z+1)/2 \leq m \leq (z+1)/2$). As an example, the kernel of an S-G
filter that fits a quadratic polynomial ($k=2$) over seven points ($z=7$) is attained here. In the first
step, $J$ is set up as follows:





$$J = \begin{bmatrix} 1 & (-3)^1 & (-3)^2 \\ 1 & (-2)^1 & (-2)^2 \\ 1 & (-1)^1 & (-1)^2 \\ 1 & (0)^1 & (0)^2 \\ 1 & (1)^1 & (1)^2 \\ 1 & (2)^1 & (2)^2 \\ 1 & (3)^1 & (3)^2 \end{bmatrix}.$$ (9)

Then, using Eq. 1, matrix $C$ is computed as Eq. 10:
$$C = \begin{bmatrix} -0.0952 & 0.1429 & 0.2857 & 0.3333 & 0.2857 & 0.1429 & -0.0952 \\ -0.1070 & -0.0714 & -0.0357 & 0 & 0.0357 & 0.0714 & 0.1071 \\ -0.0595 & 0 & -0.0357 & -0.0476 & -0.0357 & 0 & 0.0595 \end{bmatrix}.$$ (10)

The second and third rows of $C$ are the coefficients to find the filtered values' first and second
derivations at the point of interest, respectively.
**Data availability**
The synthetic database can be generated through the comprehensive steps provided here. The
Geocube measurements of Ten-mile landslide displacement are not to be publicly available.
**Author contribution**
Sohrab Sharifi: conceptualization, methodology, analysis, writing – draft preparation. Michael
Hendry: supervision, review, writing – review and editing, project administration. Renato
Macciotta: supervision, review, writing – review and editing. Trevor Evans: writing – review and
editing, validation, project administration.
**Competing interests**
The authors declare that they have no conflict of interest.
**Acknowledgment**
The authors thank CN (Canadian National Railway) for providing access to the Ten-mile site and
for purchasing the Geocube units. This research was conducted through the (Canadian) Railway





Ground Hazard Research Program, which is funded by the Natural Sciences and Engineering
Research Council of Canada (NSERC ALLRP 549684-19), Canadian Pacific Railway, CN, and
Transport Canada.

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
