# Peer review of "Evaluation of filtering methods for use on high-frequency measurements of landslide displacements"

_Natural Hazards and Earth System Sciences, 2021_

## Author Comment (AC1)

**NOTE: Reviewer's comments are in *italic* and our responses are in red letters.**

*The authors try to test three filtering methods to reduce scatter in landslide monitoring data. They check in several testing frameworks using synthetic time series whether filtering would lead to losing an early warning pulse. They also demonstrate their method with a real-world case. The article is well written and fluent. The study's motivation and implementation are of interest to the landslide research community, and they are adequate for NHESS. Before being published, I recommend some minor revisions to increase the potential impact of the research.*

The authors appreciate the efforts and thoroughness of the reviewer. Addressing the reviewer's comments has notably improved the quality of the manuscript.

*My main concern is the lack of a distinct "Discussion" section. Authors discuss their results within the section "Results and Discussion". Although they provide an overall discussion of the entire work at the end of this section, they lack linking the findings to the existing literature. Hence, I would recommend separating the "discussions" from the "results". My further minor recommendations are listed below.*

A new section dedicated to "Discussion" will be added to the revised version to highlight the link between existing literature and what has been found in this study.

*Minor comments*

*I recommend using more active sentences if possible; for example, the last sentence of the abstract could as well be written as an active sentence.*

*Introduction*

*The introduction is well written. Authors could consider extending their literature review by including some other examples. A few citations seem to appear oddly frequent. I also find series of citations after a particular statement relatively inefficient; authors should consider elaborating why they cite a specific paper (this comment applies to other sections as well).*

*Another aspect that could increase the fluency of the text is to use prefixes, for example, using "infeasible" instead of "not feasible". Would you please check other possible places where this point could be improved?*

*Lines 52–60 sound like a method, and I recommend rewriting this section a little more towards a "problem statement".*

*Methodology*

*There are a series of abbreviations used in the article, some of which make sense not to repeat each sentence. However, some of them decrease the fluency of the text, e.g., NASD or BR.*

*Line 105: "cumulative negative displacements", shouldn't it be between "0" and "-1" instead of "1" to "0", while "0" indicates the point of origin.*

*Line 106: "absolute cumulative displacements" should be from "0" to "1", not from "1" to "0", isn't it?*

*Line 134: Do the authors mean residuals when stating "ratio of scattering amplitude".*

*Lines 156–162: mentioning the GWMA acronym once might help the reader link the abbreviation to the subsection.*

*Line 169: Here, does the word "above" relate to GWMA"?*

All of the above suggestions/concerns will be addressed in the revised manuscript.

*Line 170: Authors mentioned "evenly spaced" data. Does the method also work with missing data or unevenly spaced data? Later on, in line 262, a similar issue is mentioned: when a data point is classified as an outlier, it is replaced with a new interpolated data point, which is not a proven practice. It could be an option to leave it as a NaN, or some bootstrapping and randomization process needs to be implemented.*

If points are not evenly spaced, the best solution is to adopt the regression analysis. Kernel averaging is just a simplification of S-G but the filter itself is established based on fitting curves. Explanations will be added to the revised manuscript. Regarding outliers' treatment, authors studied both linear interpolation replacement and the NaN method by adding an "omitnan" flag to the filtration script which makes the filter to overlook the NaN values. However, results were found to be approximately similar. Appropriate explanations will be added, and linear interpolation will be eliminated from the manuscript.

*Lines 181–186: I found this part of the text somewhat confusing; please consider reformulating the text.*

*Line 241: even if it is commonly used, please state the open form of GNSS.*

**Results and Discussion**

*Would it be possible to introduce sub-sections to distinguish the effects of direct and indirect filtering on the synthetic analyses?*

*Line 373: "even", is it "event"?*

All of the above suggestions/concerns will be addressed in the revised manuscript.

*Line 436: Could authors perform RMSE or another approach to quantify the trends in the data instead of visual inspection.*

The RMSE approach was not possible as this parameter is an indication of error while the "true" displacement trend, would be unknown. However, the lag quantification test on the Ten-mile landslide that presented in the latest part of section 4 serves that purpose. The values of shift ratio will be presented to clarify the explanations quantitatively.

*Line 464: Could an example of the outlier removing process be supported with a figure to visually demonstrate which type of data point authors considered an outlier.*

**Tables**

*Table 1: Could table 1 be integrated into Figure 1 to save space and better relate to one another?*

*Table 2: "60-s reading" could be mentioned as "1-m reading" to have synchrony with "1-h reading".*

*Table 4: Could this table be transformed into a figure?*

**Figures**

*Figure 4: A legend would help the reader to follow the figure as a stand-alone item. Currently, Geocubes are not explained in the figure. Resolution is also low to see the individual features correctly. Also, in subplot (a), the location of the study site is not highlighted in the map of British Columbia.*

*Figure 6: Would it be possible to combine subplots to compare them better. Y-ticks are tough to relate to at the moment.*

*Figure 9: Instead of a linear y-axis, an option could be $y^2$ (e.g., 2 4 8 16) or $y^3$ (e.g., 2 8 32 128) style access to distinguish the lines from one another for better visibility. Similarly, please consider this approach also for figure 17.*

*Figure 11: This could be shown earlier in the manuscript to understand the framework better. Similarly, figure 13 could also be placed earlier.*

*Figure 15: it resembles scenario 2 of the synthetic analyses. Consider linking them in the figure and in the main body. It might help the reader to relate synthetic analyses to the real-life case.*

*Figure 16: I found the figure and the related text in the main body somewhat confusing.*

*Figure 17: it is hard to distinguish the lines from one another.*

All of the above suggestions/concerns will be addressed in the revised manuscript.

---

## Author Comment (AC2)

**NOTE: Reviewer's comments are in *italic* and our responses are in red letters.**

*The paper "Evaluation of filtering methods for use on high frequency measurements of landslide displacements" deals with the effects of different filtering techniques to be applied to landslide displacement data in the framework of EWSs. The topic is largely relevant for the landslide community and addresses a very common problem.*

*The language is fluent and correct and the work is well designed and presented, although some improvements can be made, resulting in overall minor revisions.*

Your efforts and feedback are much appreciated. We have tried our best to address your following concerns.

*Concerning the design, while it is interesting that you have studied as many as 12 different scenarios, some of them are not very likely to represent actual landslide behaviours. In particular, I strongly recommend that you include among the scenarios a power law increase representing a tertiary creep, which is probably the most relevant trend to be detected for an EWS. Also stepped lines (that is time series characterized by cycles of seasonal activations and stabilizations) would be interesting to be studied.*

The reason of studying numerous scenarios was to make the case on differentiation between harmonic and instantaneous scenarios; otherwise, it would seem random if just one or two of each were investigated. Regarding the stepped trend, we believe that because of the versatility of the framework studied here, it is already addressed as it would be a sequential combination of the scenarios presented in our work. In numerical analysis on synthetic database, the concluding remarks are valid for new trends made by mosaicking the scenarios presented in this manuscript. As an example, by putting scenario 10 two times diagonally on top of each other as presented in the Figure 1, a stepped trend would be achieved. In Figure 1, also the filtered results after application of all three filters at BR=0.10 on unfiltered data with n/t=0.15 are shown as well. For the interest of readers, we have analyzed this specific case. In Figure 2, RMSEd of this newly generated case is presented and the same behavior as presented in the manuscript is observed. We should add that the results of this scenario were found to be independent of *n/t* ratio similar to the scenarios studied in the manuscript. The increase of error in S-G results in comparison with scenario 10 can be attributed to two incidence of pulsating effect in stepped scenario as there are two instantaneous changes.

[Figure]

*Figure 1. Stepped scenario generated using scenario 10 along with filtered results at BR=0.10 and n/t=0.15.*

[Figure]

*Figure 2. RMSEd of scenario 10 and stepped scenario*

Power law motion trend developed based on creep theory has an asymptotic behavior near the failure which based on the method presented in this paper cannot be mathematically normalized. This is true for both displacements and velocity values. As a result, the inverse-velocity diagram should be normalized which goes beyond the direct and indirect filtration methods discussed in this study. Here, only those normalized scenarios were studied that either themselves or their derivation can represent the trend and have finite values. All being said, authors would like to disclose that our intention was to establish a quantitative foundation for investigating the performance of these three filters as no such study has been dedicated to that. In this regard, the following questions are addressed: how should we select a filter bandwidth, if the application method of filters (direct or indirect) would impact results, how much the filters preserve/distort the trend, and how much lag is created when simulating real-time monitoring. Having the accumulated knowledge after answering these questions, another thorough study regarding the reliability of filters on detecting onset of acceleration moment, and forecasting failure time based on Fukuzono's method is being finalized based on the established mathematical framework in this manuscript. As stated earlier, for failing scenarios, the inverse-velocity values were normalized. The results are prepared in the form of a manuscript and will be presented to the scientific community upon publication of this paper.

*On the other hand, concerning the presentation, the results and discussions section would be better subdivided into subsections. One subdivision could be between results (objective description) and discussions (interpretations and comments). Further subsections could be added to improve readability and to separate different concepts and contents more effectively, both in the results and in the discussions sections. For example, the discussion section could be subdivided into 4 subsections, one for each filter that you analyzed, pointing out the advantages and disadvantages, and a final one to make comparisons, determine which one is better and in what circumstances and deliver the take home message. In particular, the take home message could be better highlighted, evidencing what is, in your opinion, the best filter in an operative situation. For example, in case one wants to apply Fukuzono's method, what filter would work best? And would you suggest a direct filtering on the inverse velocity (typically affected by peaks and strong variations) or an indirect filtering on velocity or even on displacement (typically presenting a power law acceleration before failure)?*

Section 4.1., which contains the results of synthetic cases analysis, will be divided into multiple sub-sections for better arrangement of remarks. Moreover, formerly Figs. 11 and 12 are now relocated to follow the $J_2$ results as referee 1 suggested this would help reader's visualization of filters' performance, and authors also agree. Additionally, a "Discussion" section will be added to summarize the main takeaways of this study.

Regarding your examples, direct and indirect filtration found to be not significant in the results of this study. However, especially for Fukuzon's method, in the follow-up study mentioned earlier, we observed on both synthetic and real-world failed cases that again how one may apply the filter (direct or indirect) did not influence the outcomes.

*Following are just few minor suggestions:*

*Line 132: "based on scaling" can be probably simplified into "by scaling".*

*309: you make reference to scenario 6 relative to fig 6 while in the caption of fig 6 there is no mention of any scenario.*

*534: please replace overstate/understate with overestimate/underestimate.*

All of the above suggestions/concerns will be addressed in the revised manuscript.

---

## Author Response (AR1)

Natural Hazards and Earth System Sciences
Subject: Submission of revised paper with reference number "nhess-2021-212"

Dear Dr. Catani,

The authors would like to thank you, all the associate editors and the reviewers for their time and valuable comments. We have carefully addressed all the comments raised by the reviewers, please find our responses in the following pages. Our responses are given in a point-by-point manner, and changes to the revised manuscript are highlighted in yellow. All the changes have been reviewed and accepted by all co-authors.

It is much hoped that the revised version is suitable to be considered for possible publication in your prestigious journal. Should you have any more concerns or questions, please do not hesitate to contact me. I am looking forward to hearing from you regarding your final decision.

Sincerely,

Sohrab Sharifi, M.Sc.
Ph.D. student
Civil and Environmental Engineering Department
University of Alberta
1-036 Natural Resources Engineering Facility
9105 116 St NW, Edmonton, AB T6G 2W2
E-mail: ssharifi@ualberta.ca
Phone: +1-(587)-589-9176

***Reviewer #1:*** We wish to express our appreciation for your precise comments. We have answered each of your points below.

***1-*** My main concern is the lack of a distinct "Discussion" section. Authors discuss their results within the section "Results and Discussion". Although they provide an overall discussion of the entire work at the end of this section, they lack linking the findings to the existing literature. Hence, I would recommend separating the "discussions" from the "results".

> ***Response to 1:*** We added a new section for "Discussion" (Page 33-35; Lines 544-580). Within this section, we tried to summarize the final takeaways for applying the studied filters.

***2-*** I recommend using more active sentences if possible; for example, the last sentence of the abstract could as well be written as an active sentence.

> ***Response to 2:*** We minimized instances of passive voice (Page 1, Lines 21-22; Page 3, Lines 64-65; Page 4, Line 88-89; Page 7, Lines 123-124 & 131-132; Page 10, Lines 183-184 & 186; Page 13, Lines 238 & 243-244; Page 24, Lines 408-409 & 412-413).

***3-*** The introduction is well written. Authors could consider extending their literature review by including some other examples. A few citations seem to appear oddly frequent. I also find series of citations after a particular statement relatively inefficient; authors should consider elaborating why they cite a specific paper (this comment applies to other sections as well).

> ***Response to 3:*** Citations were made at places where authors believe it might be of reader's interest to study further about the current subject (such as citations in the Line 100), or more evidence is needed to justify claims (such as citation in the Line 128). We elaborated on papers which we found to be aligned with the objective of the present manuscript are now elaborated more (Page 3 & 4; Lines 75-86).

**4-** Another aspect that could increase the fluency of the text is to use prefixes, for example, using "infeasible" instead of "not feasible". Would you please check other possible places where this point could be improved?

> ***Response to 4:*** Instances of such are modified now (Page 2, Lines 32 & 33; Page 3, Line 69).

**5-** Lines 52–60 sound like a method, and I recommend rewriting this section a little more towards a "problem statement".

> ***Response to 5:*** The mentioned part is now modified and parts of it which could be relocated are now moved to section 2.6 (Page 3, Lines 57-59; Page 14, Lines 258-260).

**6-** There are a series of abbreviations used in the article, some of which make sense not to repeat each sentence. However, some of them decrease the fluency of the text, e.g., NASD or BR.

> ***Response to 6:*** We minimized the unnecessary abbreviations (Page 6, Line 117; Page 10, Line 185; Page 12, Line 205; Page 13, Line 242; Page 17, Lines 315 & 316 & 317 & 319; Page 18, Line 325 & 328 & 331 & 332 & 334-335; Page 19, Lines 338 & 340 & 342 & 344 & 345 & 348 & 353 & 354 & 356; Page 20, Lines 357 & 358 & 360 & 361 & 369 & 372; Page 21, Lines 373 & 375-376 & 377 & 379 & 383; Page 22, Lines 395-396 & 401-402; Page 24, Lines 410 & 421 & 423; Page 25, Lines 425 & 426 & 432-433 & 435 & 436 & 437 & 438 & 444; Page 26, Lines 445 & 446 & 458 & 460 & 462; Page 27, Line 468; Page 28, Lines 472 & 474 & 475 & 478 & 495; Page 30, Line 507; Page 31, Lines 511 & 515; Page 32, Lines 518 & 528-529; 536-537; Page 35, Lines 591 & 593; Page 36, Lines 598 & 606 & 611 & 612 & ).

**7-** Line 105: "cumulative negative displacements", shouldn't it be between "0" and "-1" instead of "1" to "0", while "0" indicates the point of origin.

**8-** Line 106: "absolute cumulative displacements" should be from "0" to "1", not from "1" to "0", isn't it?

> ***Response to 7 & 8:*** All the interpretations made on the scenarios are valid for similar scenarios varying from 0 to $\pm 1$ and from $\pm 1$ to 0. This is due to second powers in $J_2$ and RMSE equations. However, the reviewer is correct as it might cause confusion to the reader without proper explanation. The mentioned parts are now eliminated. Correspondingly, modifications are now made and further explanations are provided (Fig 1, Line 115; Fig 2, Line 141; Page 12, Lines 206-208; Page 13, Lines 234-236).

**9-** Line 134: Do the authors mean residuals when stating "ratio of scattering amplitude".

> ***Response to 9:*** It relates to the amplitude of scatter before the filtration as one may see in the unfiltered data obtained from instruments.

**10-** Lines 156–162: mentioning the GWMA acronym once might help the reader link the abbreviation to the subsection.

> ***Response to 10:*** "Gaussian-weighted moving average" is added to the mentioned section (Page 9, Line 162).

**11-** Line 169: Here, does the word "above" relate to GWMA"?

> ***Response to 11:*** Yes, it is now added to the manuscript (Page 9, Lines 169).

**12-** Line 170: Authors mentioned "evenly spaced" data. Does the method also work with missing data or unevenly spaced data? Later on, in line 262, a similar issue is mentioned: when a data point is classified as an outlier, it is replaced with a new interpolated data point, which is not a proven practice. It could be an option to leave it as a NaN, or some bootstrapping and randomization process needs to be implemented.

**_Response to 12:_** If points are not evenly spaced, the best solution is to adopt the regression analysis. Kernel averaging is just a simplification of S-G but the filter itself is established based on fitting curves. Explanations are added (Page 10, Lines 174-178). Regarding outliers treatment, authors studied both linear interpolation replacement and NaN method by adding an "omitnan" flag to the filtration script which makes the filter to overlook the NaN values. However, results were found to be approximately similar. Appropriate explanations are added (Page 10, Line 173-176; Page 15, Line 272-273; Page 28, Line 495-496; Page 29, Line 504).

**_13-_** Lines 181–186: I found this part of the text somewhat confusing; please consider reformulating the text.

**_Response to 13:_** We added a figure to explain the difference between the two types of filtration window (Page 11, Line 188).

**_14-_** Line 241: even if it is commonly used, please state the open form of GNSS.

**_Response to 14:_** The open form of GNSS is added (Page 14, Line 248).

**_15-_** Would it be possible to introduce sub-sections to distinguish the effects of direct and indirect filtering on the synthetic analyses?

**_Response to 15:_** We divided section 4.1 to different sub-sections as suggested (Page 20, Line 370; Page 22, Line 393; Page 24, Line 407; Page 31, Line 513).

**_16-_** Line 373: "even", is it "event"?

**_Response to 16:_** It was "even" to emphasize the downside of SMA even at low bandwidth ratios. However, it is eliminated to avoid confusion (Page 19, Line 353).

**17-** Line 436: Could authors perform RMSE or another approach to quantify the trends in the data instead of visual inspection.

> ***Response to 17:*** The RMSE approach was not possible as this parameter is an indication of error while the "true" displacement trend is unknown. However, the lag quantification test on Ten-mile landslide presented in the latest part of section 4 serves that purpose. The values of shift ratio are now presented in Table 3 to clarify the explanations quantitatively (Page 33, Line 538).

**18-** Line 464: Could an example of the outlier removing process be supported with a figure to visually demonstrate which type of data point authors considered an outlier.

> ***Response to 18:*** We added a sub-figure to Fig. 18 that visually illustrates the thresholds above which is attributed to outliers (Page 31, Line 509).

**19-** Table 1: Could table 1 be integrated into Figure 1 to save space and better relate to one another?

> ***Response to 19:*** Fig. 1 is now modified as advised (Page 6, Line 115).

**20-** Table 2: "60-s reading" could be mentioned as "1-m reading" to have synchrony with "1-h reading".

> ***Response to 20:*** It is now corrected as advised (Table 1, Page 6).

**21-** Table 4: Could this table be transformed into a figure?

> ***Response to 21:*** Table 4 is replaced now with Fig. 19 (Page 33, Line 540).

**22-** Figure 4: A legend would help the reader to follow the figure as a stand-alone item. Currently, Geocubes are not explained in the figure. Resolution is also low to see the individual features correctly. Also, in subplot (a), the location of the study site is not highlighted in the map of British Columbia.

**_Response to 22:_** We added a legend and the whole figure is updated to a higher-resolution version. Ten-mile landslide is very close to Lillooet community so adding it to the Canada map inset would interfere with other texts. However, Lillooet is highlighted relative to other major cities (Page 16, Line 306).

**23-** Figure 6: Would it be possible to combine subplots to compare them better. Y-ticks are tough to relate to at the moment.

**_Response to 23:_** All subplots in Fig. 6 are now combined (Page 18, Line 320).

**24-** Figure 9: Instead of a linear y-axis, an option could be y^2 (e.g., 2 4 8 16) or y^3 (e.g., 2 8 32 128) style access to distinguish the lines from one another for better visibility. Similarly, please consider this approach also for figure 17.

**_Response to 24:_** The scales of above-mentioned figures are now modified. Also, the range of x-axis is decreased to show the variations better (Page 23, Lines 403 & 405). Formerly Fig. 17 is also modified (Page 31, Line 510).

**25-** Figure 11: This could be shown earlier in the manuscript to understand the framework better. Similarly, figure 13 could also be placed earlier.

**_Response to 25:_** The formerly Fig. 11 and Fig. 12 are now relocated in section 4.1 to follow the results of roughness factor (Page 18, Line 322). However, formerly Fig. 13 (now Fig. 14) is kept at its initial location as it is used to establish an introduction to lag quantification section. This would help the coherence of paper.

**26-** Figure 15: it resembles scenario 2 of the synthetic analyses. Consider linking them in the figure and in the main body. It might help the reader to relate synthetic analyses to the real-life case.

**_Response to 26:_** After a series of iterations, it turned out scenario 4 would resemble the trend better. An inset is now added to the figure (Page 26, Lines 456-458; Page 27, Fig. 16).

**_27-_** I found the figure and related text in the main body somewhat confusing.

**_Response to 27:_** The relevant part in the body is rewritten to clarify it better (Page 28, Lines 479 & 480-489).

**_28-_** Figure 17: it is hard to distinguish the lines from one another.

**_Response to 28:_** The mentioned figure is now updated to make lines more distinguishable from one another (Page 31, Line 510).

*Reviewer #2:* We wish to express our appreciation for your precise comments. We have answered each of your points below.

*1-* Concerning the design, while it is interesting that you have studied as many as 12 different scenarios, some of them are not very likely to represent actual landslide behaviours. In particular, I strongly recommend that you include among the scenarios a power law increase representing a tertiary creep, which is probably the most relevant trend to be detected for an EWS. Also stepped lines (that is time series characterized by cycles of seasonal activations and stabilizations) would be interesting to be studied.

> ***Response to 1:*** The reason of studying numerous scenarios was to make the case on differentiation between harmonic and instantaneous scenarios; otherwise, it would seem random if just one or two of each were investigated. Regarding the stepped trend, we believe because of the versatility of the framework studied here, it is already addressed as it would be a sequential combination of the scenarios presented in our work. In numerical analysis on synthetic database, the concluding remarks are valid for new trends made by mosaicking the scenarios presented in this manuscript. As an example, by putting scenario 10 two times diagonally on top of each other as presented in the Figure 1, a stepped trend would be achieved. In Figure 1, also the filtered results after application of all three filters at BR=0.10 on unfiltered data with $n/t$=0.15 are shown as well. We have analyzed this specific case. In Figure 2, RMSEd of this newly generated case is presented and the same behavior as presented in the manuscript is observed. We should add that the results of this scenario were found to be independent of $n/t$ ratio similar to scenarios studied in the manuscript. The increase of error in S-G results in comparison with scenario 10 can be attributed to two incidence of pulsating effect in stepped scenario as there are two instantaneous changes.

[Figure]

*Figure 1. Stepped scenario generated using scenario 10 along with filtered results at BR=0.10 and n/t=0.15.*

[Figure]

*Figure 2. RMSEd of scenario 10 and stepped scenario*

Power law motion trend developed based on creep theory has an asymptotic behavior near the failure which based on the method presented in this paper cannot be mathematically normalized. This is true for both displacements and velocity values. As a result, the inverse-velocity diagram should be normalized which goes beyond the direct and indirect filtration methods discussed in this study. Here, only those normalized scenarios were studied that either themselves or their derivation can represent the trend and have finite values. All being said, authors would like to disclose that our intention was to establish a quantitative foundation for investigating the performance of these three filters as no such study has been dedicated to that. In this regard, the following questions are addressed: how should we select a filter bandwidth, if the application method of filters (direct or indirect) would impact results, how much the filters preserve/distort the trend, how much lag is created when simulating real-time monitoring. Having the accumulated knowledge after answering these questions, another thorough study regarding the reliability of filters on detecting onset of acceleration moment, and forecasting failure time based on Fukuzono's method is being finalized based on the established mathematical framework in this manuscript. As stated earlier, for failing scenarios, the inverse-velocity values were normalized. The results are prepared in the form of a manuscript and will be presented to the scientific community upon publication of this paper.

*2-* On the other hand, concerning the presentation, the results and discussions section would be better subdivided into subsections. One subdivision could be between results (objective description) and discussions (interpretations and comments). Further subsections could be added to improve readability and to separate different concepts and contents more effectively, both in the results and in the discussions sections. For example, the discussion section could be subdivided into 4 subsections, one for each filter that you analyzed, pointing out the advantages and disadvantages, and a final one to make comparisons, determine which one is better and in what

circumstances and deliver the take home message. In particular, the take home message could be better highlighted, evidencing what is, in your opinion, the best filter in an operative situation. For example, in case one wants to apply Fukuzono's method, what filter would work best? And would you suggest a direct filtering on the inverse velocity (typically affected by peaks and strong variations) or an indirect filtering on velocity or even on displacement (typically presenting a power law acceleration before failure)?

> ***Response to 2:*** The section 4.1., which contains the results of synthetic cases analysis, are now divided to multiple sub-sections for better arrangement of remarks. Moreover, formerly Figs. 11 and 12 are now relocated to follow the $J_2$ results as referee 1 suggested this would help reader's visualization of filters' performance, and authors also agree (Page 18, Line 322; Page 20, Line 370; Page 22, Line 393; Page 24, Line 407; Page 31, Line 513). Additionally, a "Discussion" section is added to summarize the main takeaways of this study (Page 33-35; Lines 544-580).

> Regarding your examples, direct and indirect filtration found to be not significant in the results of this study. However, especially for Fukuzon's method, in the follow-up study mentioned earlier, we observed on both synthetic and real-world failed cases that again did how one may apply the filter (direct or indirect) did not influence the outcomes.

**3-**Line 132: "based on scaling" can be probably simplified into "by scaling".

> ***Response to 3:*** It is now modified as advised (Page 7, Line 133).

**4-**309: you make reference to scenario 6 relative to fig 6 while in the caption of fig 6 there is no mention of any scenario.

**_Response to 4:_** The reason was the fact that the resultant $J_2$-BR diagrams turned out to be virtually identical for the scenarios. However, to avoid confusion, it is now added to the figure's caption (Page 18, Line 321).

**5-**534: please replace overstate/understate with overestimate/underestimate.

**_Response to 5:_** These terms are now modified (Page 36, Line 617).